# Psychological distress and quality of life among Opioid Agonist Treatment service users with a history of injecting and non-injecting drug use: A cross-sectional study in Kathmandu, Nepal

Sagun Ballav Pant[1,2]*, Suraj Bahadur Thapa[1,2,3], John Howard[4], Saroj Prasad Ojha[2], Lars Lien[5,6]

1 Division of Mental Health and Addiction, Institute of Clinical Medicine, University of Oslo, Oslo, Norway, 2 Department of Psychiatry, Institute of Medicine, Tribhuvan University, Kirtipur, Nepal, 3 Division of Mental Health and Addiction, Oslo University Hospital, Oslo, Norway, 4 National Drug and Alcohol Research Centre, Faculty of Medicine, University of New South Wales, Sydney, NSW, Australia, 5 National Advisory Unit on Concurrent Substance Abuse and Mental Health Disorders, Innlandet Hospital Trust, Hamar, Norway, 6 Faculty of Social and Health Sciences, Inland Norway University of Applied Sciences, Elverum, Norway

* sagun055@gmail.com, pants@uio.no

**Data Availability Statement:** All relevant data are within the paper and its Supporting Information files.

## Abstract

### Background

Opioid use disorder is a serious public health problem in Nepal. People who use opioids often experience psychological distress and poor quality of life. Opioid agonist Treatment (OAT) is central in managing opioid dependence. This study aimed to examine factors associated with quality of life and serious psychological distress among OAT service users in the Kathmandu Valley, Nepal and compare those who had injected opioids prior to OAT and those who had not.

### Methods

A cross-sectional study with 231 was conducted using a semi-structured questionnaire, the Nepalese versions of the Kessler 6 psychological distress scale and World Health Organization Quality of Life scale (WHOQOL-BREF). Bivariate and multivariate analyses were undertaken to examine factors associated with quality of life and serious psychological distress.

### Results

Most participants were males (92%) and about half had injected opioids before initiating OAT. Serious psychological distress in the past four weeks was significantly more prevalent among participants with a history of injecting (32.2%) than those who did not inject (15.9%). In the adjusted linear regression model, those who had history of injecting were likely to have lower physical quality of life compared to non-injectors. Those self-reporting a past history of mental illness were more than seven times and those with medical comorbidity twice

**Funding:** The research is funded by the "Collaboration in Higher Education in Mental Health between Nepal and Norway- the COMENTH/NORPART project" https://www.med.uio.no/klinmed/english/research/projects/comenth/ The funding institution was not involved in data collection, analysis and manuscript writing and finalization.

**Competing interests:** The authors have declared that no competing interests exist.

more likely to have serious psychological distress over last four weeks. Lower socioeconomic status and a history of self-reported mental illness in the past were found to be significantly associated with lower quality of life on all four domains.

## Conclusion

Those who had history of injecting were younger, had frequent quit attempts, higher medical comorbidity, lower socioeconomic status and remained longer in OAT services. Alongside OAT, the complex and entangled needs of service users, especially those with a history of injecting drugs, need to be addressed to improve quality of life and lessen psychological distress.

## Background

Globally, an estimated 62 million people use opioids for non-medical reasons. This use of opioids accounted for 70 per cent of the 18 million healthy years of life lost due to disability and premature death attributed to drug use disorders in 2019. More than half of the estimated global number of opioid users reside in Asia [1]. In Nepal, opioids were the second most commonly used substance, according to the Drug Users survey-2020 [2]. Most people who use opioids in Nepal are either smoking/inhaling heroin, injecting opioids like buprenorphine and heroin, or using high doses of tramadol orally in combination with other substances [2–4]. Injecting drug users (IDU) in Nepal usually inject a combination of opioids, benzodiazepines and antihistamines called the "South Asian Cocktail", and the use of this mixture has increased health, social, economic and legal hazards in this population [5].

Opioid Agonist Treatment (OAT), primarily use of methadone, buprenorphine and buprenorphine- naloxone combination, is an evidence-based harm reduction initiative for people with opioid use disorder (OUD) that has been increasingly used in order to decrease the health, economic, and social consequences of substance use and to improve quality of life (QoL) [6]. The National Center for AIDS and STD Control (NCASC) implements OAT programs at 12 sites across 10 districts of Nepal with the help of eight Government hospitals and four Non-government organizations (NGO) where both methadone and buprenorphine are dispensed on a daily basis under the supervision of a trained health professional [7]. Nepal has been implementing a 'low threshold' OAT program, that is not exclusively for IDUs but anyone with OUD who gives informed consent and has no contraindications [8].

Several factors such as age, employment, duration of OUD, psychiatric diagnoses and psychopharmacological medication have been associated with improved health related QoL amongst those receiving OAT. However, even when improvement is observed for physical health, it has been noted that OAT service users demonstrate more psychological distress and poorer QoL compared to general population [9]. Moreover, facets of mental health and QoL are often overlooked, neglected or receive minimal attention in many harm reduction interventions [10]. This is of concern, as psychiatric co-morbidities contribute to increased mortality and morbidity among those with OUD and in OAT, with higher levels of depressive, anxiety and antisocial personality disorders being the most common [11–14]. Likewise, OUD can lower QoL via impacts on all four domains (psychological, physical, social, and emotional), [15–17] and higher levels of perceived stigma and discrimination, especially in relation to injection drug use, are also associated with higher psychological distress, unhealthy behaviors and significantly poorer QoL [18–21].

In addition to physical and mental health concerns, IDUs often have instability in many aspects of life in addition to high prevalence of infectious disease and mental disorders, such as

crime, violence, and lack of stable housing [22, 23]. Resource poor settings often face greater difficulties in addressing the diverse but entangled needs of persons with OUD, due to competing health needs and other priorities.

Nepal, was one of the first South-Asian countries to introduce harm reduction interventions for IDUs, with a needle and syringe program commencing in 1991, and OAT in 1994 [24]. However, Nepal is still not able to provide sustainable, comprehensive health and psychosocial care for people who use drugs [3]. There are relatively few studies in South-Asian settings examining psychological distress and QoL among OAT service users and its association with socio-demographic characteristics and other related variables [25–27]. The study aims to detect any differences in QoL and serious psychological distress (SPD) between OAT service users with histories of injecting and non-injecting drug use and examine the factors associated with QoL and SPD.

## Methods

### Study setting and design

A cross-sectional study among service users across five OAT sites was conducted from January, 2021 to August, 2021 in three districts in the Kathmandu valley [Kathmandu and Lalitpur two each, and Bhaktapur one]. All OAT sites providing service in Kathmandu were selected for the study. Among them, two were government hospital-based and three community based- Non-Governmental Organization run sites. The Kathmandu Valley has a population of about 2.5 million [28]. Kathmandu valley was chosen on convenience as it has the highest number of OAT sites and highest cumulative OAT service users. The center in Kathmandu represents remaining OAT sites in other cities, as all other centers are also urban based and operating as per the same treatment protocol [8].

### Sample size determination and sampling procedure

Sample size was calculated by using single population proportion formula for finite population and the selection of OAT service users based on probability sampling using simple random sampling techniques [29].

Required minimum sample size (n) = $[z^2p\ (1-p)/e^2])/\ [1+ \{z^2p\ (1-p)/e^2N\}]$

A total of 477 service users across all the OAT sites in Kathmandu valley were taking service during the study period and were used as a finite population. The proportion (p) was considered 50% in the absence of previous similar studies, with margin error at 5%, and standard normal deviation (Z) at a confidence limit of 95% [30]. The total sample size was 247 with the anticipation of 15% non-response rate to the calculated sample size.

For the required sample size of 247 OAT service users were proportionately allocated to five OAT sites and were selected randomly through computer generated random numbers after listing potential participants in each OAT site. With the response rate of 93.5%, there were 231 study OAT service users in the study (213 male and 18 female).

Eligible OAT service users were aged 18 to 60 years. Those in the first two weeks of initiation in OAT or with any organic mental disorders such as dementia, delirium and amnestic syndromes were excluded due to likely difficulties in giving a reliable history.

### Measures and instruments

A face-to-face structured interview was conducted. Information regarding the study was explained to the participants by the core study team members, and written informed consent was obtained. The Nepalese version of the Kessler-6 psychological distresses scale (K-6) and

the World Health Organization Quality of Life (WHOQOL-BREF) were used to assess psychological distress and QoL across various domains respectively. Pretesting was done with 20 service users of the estimated sample size, and the questionnaires were reviewed and revised.

**Socio-demographic questionnaire.** A semi-structured socio-demographic questionnaire was developed to assess the sociodemographic variables and OAT related information. The sociodemographic information included age, gender, education, caste, employment status, marital status, family structure, previous attempt to quit substance, presence of co-morbid medical conditions, past history of mental illness and socio-economic status (SES). SES assessment was based on the modified Kuppuswami's scale for socio-economic status [31]. Information on ethnicity was first collected based on classification of 'Caste, Ethnic and Regional Identity in Nepal', [32] which was later grouped into three categories–Brahmin and Chettris, Janajatis, Dalits and others. Co-morbid medical conditions included non-communicable diseases like tuberculosis, diabetes mellitus, hypertension and blood borne infections like HIV, Viral hepatitis (B and C), injection related thrombosis and abscess. Information on first use of illicit opioids and OAT history related information were also obtained through the questionnaire.

**World Health Organization Quality of Life (WHOQOL-BREF).** The internationally validated WHOQOL-BREF comprises 26 items in four domains: physical health (7 items), psychological health (6 items), social relationships (3 items) and environment (8 items). The two remaining WHOQOL-BREF items separately rate overall perception of QoL and overall perception of the health of an individual [33]. The WHOQOL-BREF has been validated among OAT service users [34] and translated into Nepali language and used in research in Nepal [35, 36]. It has good discriminant validity, content validity, and test-retest reliability [33, 37]. The physical health domain of WHOQOL-BREF explores activities of daily living, energy and fatigue, mobility, pain and discomfort, sleep and rest, and work capacity. The psychological domain focuses on the ability to concentrate, self-esteem, body image, spirituality and the frequency of positive or negative feelings. The social relationship domain includes personal relationship, social support, and sexual activity. The environment domain includes safety and security, home and physical environment satisfaction, financial security, health/social care availability, information and leisure activity accessibility and transportation satisfaction. The mean scores of items in each domain were used to calculate the domain score, and higher the domain score the higher the QoL.

**Kessler-6 psychological distresses scale (K-6).** The K-6 is a standardized, validated screening tool with 6 items that screen for a global measure of possible serious psychological distress (SPD) over last four weeks [38] which can be indicative of a serious mental illness [39]. Each of the following 6 questions are scored from 0 (none of the time) to 4 (all of the time): feeling nervous; hopeless; restless; that everything was an effort; was so sad that nothing could cheer him/her up; or felt worthless. Scores of 0–12 indicate not having significant psychological distress while the score of 13–24 indicate probably having serious psychological distress (SPD) over the last four weeks [38, 40]. This scale has been translated into Nepali language and used in Nepal [41, 42].

## Statistical analysis

Data was entered in SPSS version 27 for data analysis [43]. Descriptive statistics from the data such as mean and standard deviations were calculated for the continuous variables and absolute numbers and percentages for the categorical variables. The normal distribution of the continuous variables was checked by using visual inspection, assessment of skewness and kurtosis and Kolmogorov-Smirnov test [44]. Bivariate analyses were done using Chi square and

independent sample t tests, which were used to compare categorical variables and means for continuous variables, respectively. The Mann-Whitney U-test was used for not- normally distributed continuous variable. Multivariate analyses were performed with linear and logistic regression models. SPD as a dependent variable was categorized as binary into no and yes (having SPD) included in the final logistic model, where history of use of injectables, age, gender, education, employment, marital status, past history of self-reported mental illness, previous substance quit attempt, being in custody after OAT enrollment, medical co-morbidity, ethnicity, SES and duration since OAT enrollment were kept as dependent variables. The dependent variables were selected based on their statistical and clinical significance. For the liner regression, QoL continuous scores (in four domains- physical, psychological, environmental and social) were kept as dependent variable and the same factors as above were retained as independent variables. Standardized Beta was reported for linear and adjusted odds ratio for logistic regression. For both liner and logistic regression model independent variables were checked for confounding and those only with variation inflation factor less than 2 were included in the final model. The level of significance for all statistical analysis was set at p< 0.05.

## Ethical consideration

Ethical approval for this study was obtained from the ethical review board of the Nepal Health Research Council (Ref. no: 1698). Additional ethical approval was obtained from Regional Committees for Medical and Health Research Ethics in Norway (Ref. no: 154194). Permission was also obtained from each of the OAT sites and written informed consent from the study participants.

## Results

Among the 231 service users, 113 had never used opioids via injection (non-IDU) and 118 injected opioids before enrollment into OAT. As can be seen in Table 1, there was a significant difference for age, with mean age of non-IDU service users being 32.1± 6.3 years and 35.3 ± 7.9 years for the IDU. There were few women in either group and non-IDU were better educated. Significantly more IDU were separated or divorced and less than half non-IDU had attempted to quit substance use before OAT initiation. About one- third of non-IDU had co-morbid medical conditions which was significantly less for IDUs, and had higher SES.

Methadone was the most used current OAT modality compared to buprenorphine among both groups. Less non-IDU service users were arrested or taken into police custody after OAT enrollment than IDU. The median duration of OAT use was significantly less for non- IDU service users. (Table 2).

Statistically significant differences can be observed in Table 3 for all QoL domains and the overall QoL with lower mean scores for all four domains in IDU compared to non-IDU and overall QoL. The highest mean difference was observed in the environmental domain. Likewise, presence of SPD over last four weeks was higher among IDUs and the median K-6 score was significantly lower for non-IDUs.

As seen in Table 4, using the adjusted logistic regression model, IDU status did not show significant association with the SPD. Those having history of self- reported mental illness in the past were more than seven times more likely to have SPD within the last four weeks. Likewise, those with history of medical comorbidity also were around 2.28 times more like to have SPD within the last four weeks than those who did not.

IDU were likely to have lower physical QoL compared to non-IDU as seen in Table 5. Age was also positively associated with better physical and social QoL. A history of self-reported

**Table 1. Socio-demographic characteristics of non-injecting (non-IDU) and injecting drug users (IDU).**

| Characteristics | Non- IDU | | IDU | | p-value |
|---|---|---|---|---|---|
| N = 231 | n | % | n | % | |
| *Age in years, mean ± SD | 32.1 ± 6.3 | | 35.3 ± 7.9 | | 0.001 |
| **Gender** | | | | | |
| Male | 102 | 90.3 | 111 | 94.1 | 0.405 |
| Female | 11 | 9.7 | 7 | 5.9 | |
| **Education** | | | | | |
| Primary and lower | 7 | 6.2 | 16 | 13.6 | 0.021 |
| Secondary | 72 | 63.7 | 82 | 69.5 | |
| University and above | 34 | 30.1 | 20 | 16.9 | |
| **Ethnicity[a]** | | | | | |
| Brahmin/Chhetri | 46 | 40.7 | 40 | 33.9 | 0.752 |
| Janajati | 59 | 52.2 | 69 | 58.5 | |
| Dalit and others | 8 | 7.1 | 9 | 7.6 | |
| **Employment status** | | | | | |
| Employed | 32 | 28.3 | 41 | 34.8 | 0.216 |
| Unemployed | 25 | 22.1 | 32 | 27.1 | |
| Self-employed | 56 | 49.6 | 45 | 38.1 | |
| **Marital status** | | | | | |
| Married | 64 | 56.6 | 64 | 54.2 | 0.004 |
| Unmarried | 45 | 39.8 | 35 | 29.7 | |
| Separated/ divorced | 4 | 3.6 | 19 | 16.1 | |
| **Types of family** | | | | | |
| Nuclear | 54 | 47.8 | 65 | 55.1 | 0.328 |
| Extended | 59 | 52.2 | 53 | 44.9 | |
| **Previous attempt to quit substance use** | | | | | |
| Yes | 49 | 43.4 | 88 | 74.6 | 0.001 |
| No | 64 | 56.6 | 30 | 25.4 | |
| **Presence of Co-morbid medical conditions** | | | | | |
| Yes | 35 | 32.4 | 73 | 67.6 | 0.001 |
| No | 78 | 63.4 | 45 | 36.6 | |
| **Past history of self-reported mental illness** | | | | | |
| Yes | 15 | 13.3 | 23 | 19.5 | 0.273 |
| No | 98 | 86.7 | 95 | 80.5 | |
| **SES** | | | | | |
| Upper class | 70 | 61.9 | 54 | 45.8 | 0.014 |
| Lower class | 43 | 38.1 | 64 | 54.2 | |

Footnote:

*Independent sample t-tests for continuous variables and chi-square test for categorical variables

[a]-Brahmin and Chhetri are the highest ethnic group, Janajati are an indigenous group, while Dalits are underprivileged, lowest ethnicity in Nepal

SD: Standard Deviation

mental illness in the past was associated with lower QoL on all four domains of QoL and previous quit attempt was also associated with lower QoL on physical, psychological and environmental domains.

Likewise, those with upper SES were more likely to have a better QoL compared to those with lower SES with higher QoL score across all four domains of QoL.

**Table 2. Comparison of Opioid Agonist Treatment related characteristics between non-injecting and injecting drug users.**

| Characteristics | Non-IDU | | IDU | | p-value |
|---|---|---|---|---|---|
| N = 231 | n | % | n | % | |
| **Current OAT modality** | | | | | |
| Methadone | 91 | 80.5 | 82 | 69.5 | 0.053 |
| Buprenorphine | 22 | 19.5 | 36 | 30.5 | |
| **Provision of 'take away' OAT (Last month)** | | | | | |
| Yes | 21 | 18.6 | 22 | 18.6 | 0.991 |
| No | 92 | 81.4 | 96 | 81.4 | |
| **Arrested or taken into custody after OAT enrollment** | | | | | |
| Yes | 10 | 8.8 | 31 | 26.3 | 0.001 |
| No | 103 | 91.2 | 87 | 73.7 | |
| | n | Median (IQR) | n | Median (IQR) | |
| Duration since OAT enrollment* (months) | 113 | 7 (2–18) | 118 | 24 (10–60) | 0.001 |

Footnote:

* Mann-Whitney U-test for continuous variables and chi-square test for categorical variables

SD: Standard Deviation

IQR: Interquartile range

## Discussion

This study appears to be one of the first that explored and compared psychological distress and QoL among OAT service users with injecting and non-injecting histories. It is well recognized that QoL and psychological distress are influenced by multiple and entangled factors which can exacerbate each other and produce further complexity. Some arise from genetic predispositions associated with physical and mental health outcomes, others are associated with social and structural determinants of health [45].

Non-IDUs in this study were slightly younger compared to IDUs but the mean age of both groups was similar to people with OUD in India. Similarly, service users were predominantly

**Table 3. Difference in Quality of Life (QoL) and psychological distress among non IDU and IDUs.**

| Variables (N = 231) | Non IDU | | IDU | | |
|---|---|---|---|---|---|
| | Mean | SD | Mean | SD | p-value |
| Physical QoL* | 26.8 | 4.5 | 24 | 4.7 | 0.001 |
| Psychological QoL | 22 | 3.6 | 20.1 | 4.8 | 0.001 |
| Social QoL | 10.4 | 2.6 | 9.3 | 2.9 | 0.004 |
| Environmental QoL | 28.4 | 3.9 | 27 | 5.1 | 0.014 |
| Overall QoL | 94.4 | 12.2 | 86.7 | 16.2 | 0.001 |
| | n | % | n | % | |
| No Serious psychological distress (SPD) | 95 | 84.1% | 80 | 67.8% | 0.004 |
| Serious psychological distress (SPD) | 18 | 15.9% | 38 | 32.2% | |
| | n | Median (IQR) | n | Median (IQR) | |
| K-6 score** | 113 | 2 (0–7) | 118 | 5 (0–14) | 0.005 |

Footnote:

*Independent sample t-tests for continuous variables and chi-square test for categorical variables

**Mann-Whitney U-test

SD: Standard Deviation

IQR: Interquartile range

**Table 4. Factors associated with Serious psychological distress (SPD).**

| Variables (N = 231) | Unadjusted OR (95% CI) | Adjusted OR (95% CI) |
|---|---|---|
| **IDU (vs. Non-IDU)** | | |
| Yes | 2.51 (1.32, 4.73) ** | 2.02 (0.89, 4.6) |
| **Age** | 1.01 (0.97, 1.05) | 0.98 (0.92, 1.04) |
| **Gender (vs. Male)** | | |
| Female | 1.63 (0.58, 4.66) | 2.1 (0.56, 7.9) |
| **Education (vs. Primary and lower)** | | |
| Secondary | 0.87 (0.46, 1.64) | 0.76 (0.23, 2.51) |
| University and above | 0.99 (0.49, 2.01) | 0.68 (0.15,3.02) |
| **Employment (vs. Employed)** | | |
| Unemployed | 1.93 (0.45, 8.32) | 2.13(0.35, 13.01) |
| Self-employed | 1.40 (0.76, 2.60) | 1.1 (0.52, 2.28) |
| **Marital status (vs. Married)** | | |
| Unmarried | 0.86 (0.46, 1.64) | 0.67 (0.28, 1.61) |
| Separated/ divorced/ widowed | 0.86 (0.30, 2.42) | 0.46 (0.13, 1.76) |
| **Family type (vs. Nuclear)** | | |
| Extended | 0.92 (0.51, 1.69) | 1.1 (0.53, 2.30) |
| **Ethnicity (vs. Brahmin and Chettri)** | | |
| Janajati | 0.91 (0.5,1.66) | 0.66 (0.32,1.4) |
| Dalit and others | 0.65 (0.18–2.4) | 0.36 (0.08,1.64) |
| **Past history of self-reported mental illness (vs. No)** | | |
| Yes | 6.43 (3.06, 13.52) *** | 7.5 (3.15, 17.85) *** |
| **Tried quitting substance use (vs. no)** | | |
| Yes | 1.62 (0.86, 3.06) | 0.99 (0.46, 2.12) |
| **Been in custody after enrollment in OAT (vs no)** | | |
| Yes | 1.83 (0.88, 3.79) | 1.19 (0.49, 2.9) |
| **History of comorbidity (vs. no)** | | |
| Yes | 2.85 (1.52, 5.36) *** | 2.28 (1.04, 4.97) * |
| **SES (vs. Lower class)** | | |
| Upper class | 1.35 (0.76, 2.43) | 0.81 (0.36, 1.82) |
| **Duration since OAT enrollment** | 1.01 (1,1.01) | 1.00 (0.99,1.01) |

Footnote:

* p-value <0.05

** p-value <0.01

*** p-value < 0.001

OR: Odds Ratio

CI: Confidence interval In the linear regression models for different domains of QoL

male as in many countries, including neighboring India which shares cultural similarities [19, 25]. The possible reasons for the gender differences may be due to higher level of perceived stigma, and barriers in access to health care facilities and treatment including OAT for females [46]. As expected, IDUs had significantly more medical co-morbidities, lower SES and were more commonly taken into police custody despite being in OAT. Similar observations have been observed in developed settings [47].

In this study the estimated prevalence rate of SPD among non IDUs and IDUs were lower compared to global scenario [46] and regional studies form Asia [48, 49]. A study among male IDUs from needle and syringe program in Delhi, India found extremely high rates of

**Table 5. Factors associated with Quality of life (QoL).**

| | Physical QOL | Psychological QOL | Social relationships QOL | Environmental QOL |
|---|---|---|---|---|
| Variables (N = 231) | Beta (95% CI) | Beta (95% CI) | Beta (95% CI) | Beta (95% CI) |
| **IDU (vs. non-IDU)** | | | | |
| Yes | -0.14 (-3.00, -0.34) * | -0.09 (-2.46, 0.91) | -0.07 (-1.48, 0.75) | -0.02 (-1.96, 1.68) |
| **Age** | 0.17 (0.01, 0.21) * | 0.14 (-0.02, 0.56) | 0.19 (0.02, 0.13) * | 0.08 (-0.50, 0.15) |
| **Gender (vs. Male)** | | | | |
| Female | -0.07 (-3.54, 0.92) | -0.08 (-3.33, 0.75) | -0.06 (-1.97, 0.74) | -0.05 (-3.09, 1.33) |
| **Education (vs. Primary and lower)** | | | | |
| Secondary | 0.20 (-0.06, 4.04) | 0.19 (-0.19, 3.56) | 0.07 (-0.81, 1.67) | 0.21 (0.02, 4.09) |
| University and above | 0.07 (-1.62, 3.31) | 0.12 (-0.97, 3.54) | -0.03 (-1.69, 1.3) | 0.20 (-0.26, 4.62) |
| **Employment (vs. Employed)** | | | | |
| Unemployed | -0.09 (-5.59, 0.65) | -0.19 (-5.37, 0.33) | 0.04 (-1.32, 2.45) | -0.09 (-5.27, 0.83) |
| Self-employed | -0.06 (-1.75, 0.66) | -0.05 (-1.51, 0.69) | -0.07 (-1.14, 0.32) | -0.05 (-1.63, 0.76) |
| **Marital status (vs. Married)** | | | | |
| Unmarried | 0.07 (-0.67, 2.08) | 0.01 (0.01, 0.14) | 0.01 (-0.77, 0.89) | 0.04 (-1.02, 1.70) |
| Separated/ divorced/ widowed | 0.02 (-1.70, 2.30) | 0.07 (-0.80, 2.81) | -0.07 (-1.9, 0.52) | 0.05 (-1.16, 2.81) |
| **Family type (vs. Nuclear)** | | | | |
| Extended | 0.06 (-0.57, 1.80) | 0.09 (-0.32, 1.85) | 0.02 (-0.59, 0.83) | 0.10 (-0.26, 2.10) |
| **Ethnicity (vs. Brahmin and Chettri)** | | | | |
| Janajati | -0.10 (-2.13, 0.24) | -0.02 (-1.26, 0.91) | 0.05 (-0.42, 1.01) | -0.06 (-1.69, 0.66) |
| Dalit and others | -0.01 (-2.43, 2.14) | 0.09 (-0.58, 3.60) | 0.09 (-0.39, 2.38) | -0.01 (-2.41, 2.12) |
| **Past history of self-reported mental illness (vs. No)** | | | | |
| Yes | -0.30 (-5.42, -2.30) *** | -0.23 (-4.12, -1.27) *** | -0.26 (-2.94, -1.05) *** | -0.21 (-4.19, -1.09) *** |
| **Tried quitting substance use (vs. no)** | | | | |
| Yes | -0.15 (-2.67, -2.50) * | -0.18 (-2.65, -0.43) * | -0.12 (-1.43, -0.04) | -0.14 (-2.49, -0.09) * |
| **Been in custody after enrollment in OAT (vs no)** | | | | |
| Yes | -0.05 (-2.22, 0.89) | -0.01 (-1.51, 1.33) | -0.02 (-1.12, 0.76) | -0.03 (-1.87, 1.21) |
| **History of comorbidity (vs. no)** | | | | |
| Yes | -0.05 (-1.77, 0.75) | -0.12 (-2.17, 0.14) | -0.10(-1.33, 0.19) | -0.02 (-1.43, 1.07) |
| **SES (vs. Lower class)** | | | | |
| Upper class | 0.14 (0.08, 2.62) * | 0.20 (0.56, 2.92) ** | 0.28 (0.78, 2.35) *** | 0.31 (1.51, 4.08) *** |
| **Duration since OAT enrollment** | -0.89 (-0.03, 0.01) | -0.11 (-0.03, 0.01) | -0.02 (-0.01, 0.01) | 0.01 (-0.02, 0.02) |

Footnote:

*p-value <0.05

** p-value <0.01

*** p-value < 0.001

CI: Confidence interval

participants with depressive (84%) and anxiety (71%). Their study population was considerably more socially disadvantaged than the present study, with high proportions being illiterate, homeless and living on a very low income/being dependent on scavenging [50].

The current study demonstrated that despite many months in OAT, both IDU and non-IDUs showed high SPD over last four weeks. Left untreated, the risk of poor health outcomes, of relapse after treatment, workplace productivity loss and even premature mortality [51].

The study also examined the pattern of impairments in different QoL domains among OAT service users and explored the relationship between different sociodemographic and clinical variables and various domains of QoL. The findings from previous studies showed that

heroin-dependent participants had poorer QoL than controls in the general well-being items, physical, psychological, environmental and social relationship domains and total WHOQoL scores [16, 52–54]. However, the current study demonstrate that QoL scores were significantly lower in the IDU group across all the domains in a bivariate analysis, but on multivariate analysis this was only significant for the physical domain. This may be due to the higher level of physical health comorbidity in our sample.

Given the importance of QoL and psychological distress among both injecting and non-injecting users, the findings demonstrate that QoL was lower in the IDU group compared to non-IDUs in the physical domain and IDUs had spent more time in OAT compared to non-IDUs. With the passage of time, it was expected that the benefits accumulated overtime would lead to better QoL [26]. Despite this, IDU fared poorly on most domains of QoL.

Past attempts to quit substance use was associated with poor QoL. While this was an interesting observation, no similar studies were identified. It is well known that OUD has high prevalence of relapse and previous research has shown that stressful life situations, negative mood persisting over time and failed attempts at quitting substance use may reinforce a sense of 'failure' and hence negatively impact QoL [55].

A self-reported past history of mental illness was associated with lower QoL in all four domains in IDU (physical, psychological, social and environment). Studies in South Asia and the Middle East show high lifetime prevalence of psychiatric disorders, especially depression and anxiety, psychological distress and lower QoL among opioid users [12, 15, 56–62], consistent with numerous studies from Europe, Nepal, Australia, the US, Slovakia, Taiwan and Vietnam [48, 63–68]. Effectively addressing mental health issues is crucial, and the studies reveal that QoL of people with opioid dependence improves with OAT and provision of interventions addressing their individual and complex needs [25, 54].

In the current study, the upper SES OAT service users had better QoL than those with lower SES, highlighting well recognized health inequalities. The Scott et al. study demonstrated substantially lower personal well-being and related psychological stress and SES over time in a population of IDUs, but that housing and health services could make a difference [64]. Further research might explore potentially direct and indirect impacts of a broader range of social and structural variables that appear to be associated with QoL and SPD, such as the role of perceived stigma and self-stigma [20, 21, 69]. This research would benefit from quantitative, qualitative, longitudinal, and intervention studies.

## Implications

Much of what is canvassed above is not new to OAT services, but how to respond to build optimism, resilience, and agency in service users to improve QoL and reduce psychological distress is challenging, especially in resource poor settings. Sites of OAT programs must be more than a 'clinic' where service users merely attend, get dosed, have minimal contact with staff, and depart. Ideally, OAT staffing with a diversity of professional and ex-consumer/peer could better respond to this complexity; such as psychiatrists, nurses, psychologists, social workers, occupational therapists, peer educators. This can however have funding implications in low resource settings. Consequently, the development of functional ongoing links to appropriate and accessible ancillary services would be essential, guided and promoted via policy reforms.

## Strengths and limitations

This study provides novel evidence accentuating the distinct needs of people on OAT who injected opioids. Reliability of data was assured as data were collected using standardized and validated tools/instruments. However, there are some limitations. The cross-sectional study

design limits establishing causation when considering the association between SPD and QoL and factors associated with them. The study might be subjected to recall bias as some of the questions depend on subjective memory. Additionally, the impact of opioid use at baseline was not assessed. The study was undertaken in OAT sites from Kathmandu Valley; hence it might be difficult to generalize findings to all service users in OAT in Nepal or elsewhere. However, it can provide some guidance for further research to inform policy and guidelines for the development of evidence-informed interventions for OAT programs.

## Conclusion

Service users in OAT programs in this study reported high levels of SPD within last four weeks and low QoL especially among IDUs. A history of self-reported past mental illness affected both SPD and QoL. It is well recognized that more than pharmacotherapy is required to better meet the multiple, entangled and complex needs of OAT service users, and focus on health, wellbeing and quality life would assist. To identify more clearly key and modifiable contributing factors, and which approaches and interventions are the most efficient and effective in buffering the onerous impacts of psychological distress and low QoL particularly among people who inject drugs in resource-poor settings.

## Supporting information

**S1 Dataset.**
(SAV)

## Acknowledgments

We would like to express our gratitude to all the OAT sites of the Kathmandu valley. We are very grateful to Dr. Rolina Dhital and Dr. Richa Shah for statistical inputs and Mr. Suvash Nayaju and Dr. Dipesh Bhattarai for reviewing and proof reading the manuscript.

## Author Contributions

**Conceptualization:** Sagun Ballav Pant.

**Data curation:** Sagun Ballav Pant.

**Formal analysis:** Sagun Ballav Pant, Suraj Bahadur Thapa, Lars Lien.

**Funding acquisition:** Suraj Bahadur Thapa.

**Methodology:** Sagun Ballav Pant, John Howard.

**Project administration:** Sagun Ballav Pant, John Howard.

**Resources:** Suraj Bahadur Thapa, John Howard, Lars Lien.

**Software:** Suraj Bahadur Thapa, John Howard, Lars Lien.

**Supervision:** Suraj Bahadur Thapa, Saroj Prasad Ojha, Lars Lien.

**Validation:** Suraj Bahadur Thapa, Saroj Prasad Ojha, Lars Lien.

**Visualization:** Sagun Ballav Pant, Suraj Bahadur Thapa, Lars Lien.

**Writing – original draft:** Sagun Ballav Pant.

**Writing – review & editing:** Suraj Bahadur Thapa, John Howard, Saroj Prasad Ojha, Lars Lien.

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
