## [Decision Letter · Decision Letter 0]

12 Dec 2022

PONE-D-22-23448Psychological distress and quality of life among Opioid Agonist Treatment service users with histories of injecting and non-injecting: A cross-sectional study in Kathmandu, NepalPLOS ONE

Dear Dr. Pant,

Thank you for submitting your manuscript to PLOS ONE. After careful consideration, we feel that it has merit but does not fully meet PLOS ONE’s publication criteria as it currently stands. Therefore, we invite you to submit a revised version of the manuscript that addresses the points raised during the review process.

Both Reviewers find that the work is original and well written in general terms. However, a more concise results section is needed, and the execution of the Kolmogorov Smirnoff test is necessary to decide whether to proceed with parametric or not parametric analyses.Based on the Reviewers' suggestions, the discussion should also be amended, to provided thorough evidence-based  explanations of the results.

We look forward to receiving your revised manuscript.

Kind regards,

Eleni Petkari

Academic Editor

PLOS ONE

Journal Requirements:

Reviewers' comments:

Reviewer's Responses to Questions

**Comments to the Author**

1. Is the manuscript technically sound, and do the data support the conclusions?

Reviewer #1: Partly

Reviewer #2: Yes

2. Has the statistical analysis been performed appropriately and rigorously? 

Reviewer #1: Yes

Reviewer #2: Yes

3. Have the authors made all data underlying the findings in their manuscript fully available?

Reviewer #1: Yes

Reviewer #2: Yes

4. Is the manuscript presented in an intelligible fashion and written in standard English?

Reviewer #1: Yes

Reviewer #2: Yes

5. Review Comments to the Author

Reviewer #1: The study entitled “Psychological distress and quality of life among Opioid Agonist Treatment service users with histories of injecting and non-injecting: A cross-sectional study in Kathmandu, Nepal” can be improved with the following considerations.

1. Briefly describe different types of Opioid Agonist Treatment (OAT) in the Introduction, such as Methadone Maintenance Treatment (MMT).

2. The statement “OUD can lower QoL via impacts on all four domains (psychological, physical, social, and emotional), [14] and higher levels of perceived stigma and discrimination, especially in relation to injection drug use, are also associated with higher psychological distress, unhealthy behaviors and significantly poorer QoL [15,16]” can also be supported by the following relevant publications.

Lin, C.-Y., Chang, K.-C., Wang, J.-D., & Lee, L. J.-H. (2016). Quality of life and its determinants of heroin addicts receiving methadone maintenance program: comparison with matched referents from general population. Journal of the Formosan Medical Association, 115(9), 714-727.

Chang, K.-C., & Lin, C.-Y. (2015). Effects of publicly-funded and quality of life on attendance rate among methadone maintenance treatment patients in Taiwan: an 18-month follow-up study. Harm Reduction Journal, 12, 40.

Cheng, C.-M., Chang, C.-C., Wang, J.-D., Chang, K.-C., Ting, S.-Y., & Lin, C.-Y. (2019). Negative impacts of self-stigma on the quality of life of patients in methadone maintenance treatment: the mediated roles of psychological distress and social functioning. International Journal of Environmental Research and Public Health, 16, 1299.

Saffari, M., Chen, H.-P., Chang, C.-W., Fan, C.-W., Huang, S.-W., Chen, J.-S., Chang, K.-C., & Lin, C.-Y. (2022). Does sleep quality mediate the associations between problematic internet use and quality of life in people with substance use disorder? BJPsych Open, 8, e155.

Chang, C.-W., Chang, K.-C., Griffiths, M. D., Chang, C.-C., Lin, C.-Y., Pakpour, A. H. (2022). The mediating role of perceived social support in the relationship between perceived stigma and depression among individuals diagnosed with substance use disorders. Journal of Psychiatric and Mental Health Nursing, 29(2), 307-316.

Chang, C.-C., Chang, K.-C., Hou, W.-L., Yen, C.-F., Lin, C.-Y., & Potenza, M. N. (2020). Measurement invariance and psychometric properties of Perceived Stigma toward People who use Substances (PSPS) among three types of substance use disorders: heroin, amphetamine, and alcohol. Drug and Alcohol Dependence, 216, 108319.

Chang, K.-C., Lin, C.-Y., Chang, C.-C., Ting, S.-Y., Cheng, C.-M., & Wang, J.-D. (2019). Psychological distress mediated the effects of self-stigma, psychological distress, and quality of life in opioid-dependent individuals. Plos One, 14(2), e0211033.

3. When introduction the WHOQOL-BREF (i.e., line 174), please also mention that the WHOQOL-BREF has been validated among heroin users.

Chang, K.-C., Wang, J.-D., Tang, H.-P., Cheng, C.-M., & Lin, C.-Y. (2014). Psychometric evaluation using Rasch analysis of the WHOQOL-BREF in heroin-dependent people undergoing methadone maintenance treatment: further item validation. Health and Quality of Life Outcomes, 12, 148.

4. Please do not use p=<0.001. This presentation is confusing as no one know if it means p=0.001 or p<0.001.

5. The Results section is lengthy and has redundancy between text and tables. The authors are suggested to report concise results in text and refer the readers to read the details in tables.

6. The authors have defined QoL as quality of life. Then, they should adhere to using this abbreviation. However, the authors sometimes still use quality of life (e.g., line 316; quality of life among OAT service users).

7. The Discussion may discuss the QoL and psychological distress issues among people with OAT are related to stigma among this population. The authors may consult the references I mentioned earlier. The authors may also encourage future studies to know the stigma levels (including perceived stigma and self-stigma) among this population given that valid instruments have been developed (e.g., Chang et al., 2020; Fan et al., 2022).

Chang, C.-C., Chang, K.-C., Hou, W.-L., Yen, C.-F., Lin, C.-Y., & Potenza, M. N. (2020). Measurement invariance and psychometric properties of Perceived Stigma toward People who use Substances (PSPS) among three types of substance use disorders: heroin, amphetamine, and alcohol. Drug and Alcohol Dependence, 216, 108319.

Fan, C.-W., Chang, K.-C., Lee, K.-Y., Yang, W.-C., Pakpour, A. H., Potenza, M. N., & Lin, C.-Y. (2022). Rasch Modeling and Differential Item Functioning of the Self-Stigma Scale-Short Version Among People with three different Psychiatric Disorders. International Journal of Environmental Research and Public Health, 19, 8843.

Reviewer #2: Psychological distress and quality of life among Opioid Agonist Treatment service users with histories of injecting and non-injecting: A cross-sectional study in Kathmandu, Nepal

Thank you for the opportunity to review this paper. The manuscript is well written, the topic is interesting, and the results provide important information for improving the effectiveness of the OAT program, but there is still the need to address some shortcomings.

Specific comments:

- In the title, emphasize that it is about drug use when it comes to the division of participants into injecting and non-injecting “Psychological distress and quality of life among Opioid Agonist Treatment service users with a history of injecting and non-injecting drug use: A cross-sectional study in Kathmandu, Nepal”

- In the introduction, the problem is nicely and chronologically explained, but the introduction should be a little more concise or shorter.

- In the methodology, it is necessary to briefly explain why you choose 5 out of the 12 cities where the OAT programs were implemented, and whether the programs in those cities differ from the others in terms of content. This is important because of the external validity of the study and the generalizability of the results.

- It is not recommended to start sentences with abbreviations

- Have you used the Kolmogorov - Smirnov test to test the normality of the data distribution, in case the data distribution is not normal it is necessary to use non-parametric tests.

- Start the results with a description of the sample, not with the sentence "Table 1 summarizes the socio-demographic characteristics of the OAT service user……". This sentence should be placed after the textual description of the contents of Table 1.

- In the tables, it is necessary to express one p-value for comparing the value of one independent categorical variable between injecting and non-injecting drug users, for example, for the variable education we have three p-values instead of one. It is necessary to check that the same error is not repeated in the tables below. P value is expressed to three decimal places, and values less than 0.001, e.g. 0.000 is displayed as 0.001

- In line 231, explain the abbreviation SD.

- In line 240 the sentence "Regarding SES 70 (61.9%) belonged to upper SES as 241 compared to 54(45.8%) non IDU (p=0.014)" is not clear.

- A cross-sectional study design does not allow the use of words such as predictor or risk factor because we cannot prove causality, but instead the construct "factor associated with" can be used.

- In line 325 “The possible reasons for the gender differences may be due to higher level of perceived stigma, and barriers in access to health care facilities and treatment including OAT for females”…. Are there other gender differences that could be the cause of the obtained results, maybe differences in employment, income...what do other studies say.

- In the discussion, the data from the results are often repeated, the discussion should be based more on the comparison with the results from other studies for a potential explanation of them.

6. PLOS authors have the option to publish the peer review history of their article (what does this mean?). If published, this will include your full peer review and any attached files.

Reviewer #1: No

Reviewer #2: No

---

## [Author Response · Author response to Decision Letter 0]

16 Jan 2023

Reviewer #1:

Comment 1: Briefly describe different types of Opioid Agonist Treatment (OAT) in the Introduction, such as Methadone Maintenance Treatment (MMT).

Our response: Thank you for your comment. Opioid Agonist Treatment (OAT) includes taking opioid agonist medications like methadone, buprenorphine and buprenorphine- naloxone combination. 

We have added a sentence mentioning different types of OAT modalities in line 83-84, Page 4 “Opioid Agonist Treatment (OAT), primarily use of methadone, buprenorphine and buprenorphine- naloxone combination, is an evidence-based harm reduction initiative.”

Additionally, we also thought the reviewer wanted us to mention OAT modalities used in Nepal, for that we have mentioned that both methadone and buprenorphine are provided on a daily basis by a trained health professional. 

Line 87-91, page 4,now reads as:

“The National Center for AIDS and STD Control (NCASC) implements…….. Non-government organizations (NGO) where both methadone and buprenorphine are dispensed on a daily basis under the supervision of a trained health professional.”

Comment 2: The statement “OUD can lower QoL via impacts on all four domains (psychological, physical, social, and emotional), [14] and higher levels of perceived stigma and discrimination, especially in relation to injection drug use, are also associated with higher psychological distress, unhealthy behaviors and significantly poorer QoL [15,16]” can also be supported by the following relevant publications.

Lin, C.-Y., Chang, K.-C., Wang, J.-D., & Lee, L. J.-H. (2016). Quality of life and its determinants of heroin addicts receiving methadone maintenance program: comparison with matched referents from general population. Journal of the Formosan Medical Association, 115(9), 714-727.

Chang, K.-C., & Lin, C.-Y. (2015). Effects of publicly-funded and quality of life on attendance rate among methadone maintenance treatment patients in Taiwan: an 18-month follow-up study. Harm Reduction Journal, 12, 40.

Cheng, C.-M., Chang, C.-C., Wang, J.-D., Chang, K.-C., Ting, S.-Y., & Lin, C.-Y. (2019). Negative impacts of self-stigma on the quality of life of patients in methadone maintenance treatment: the mediated roles of psychological distress and social functioning. International Journal of Environmental Research and Public Health, 16, 1299.

Saffari, M., Chen, H.-P., Chang, C.-W., Fan, C.-W., Huang, S.-W., Chen, J.-S., Chang, K.-C., & Lin, C.-Y. (2022). Does sleep quality mediate the associations between problematic internet use and quality of life in people with substance use disorder? BJPsych Open, 8, e155.

Chang, C.-W., Chang, K.-C., Griffiths, M. D., Chang, C.-C., Lin, C.-Y., Pakpour, A. H. (2022). The mediating role of perceived social support in the relationship between perceived stigma and depression among individuals diagnosed with substance use disorders. Journal of Psychiatric and Mental Health Nursing, 29(2), 307-316.

Chang, C.-C., Chang, K.-C., Hou, W.-L., Yen, C.-F., Lin, C.-Y., & Potenza, M. N. (2020). Measurement invariance and psychometric properties of Perceived Stigma toward People who use Substances (PSPS) among three types of substance use disorders: heroin, amphetamine, and alcohol. Drug and Alcohol Dependence, 216, 108319.

Chang, K.-C., Lin, C.-Y., Chang, C.-C., Ting, S.-Y., Cheng, C.-M., & Wang, J.-D. (2019). Psychological distress mediated the effects of self-stigma, psychological distress, and quality of life in opioid-dependent individuals. Plos One, 14(2), e0211033.

Our response: Thank you for the suggestions of excellent papers. We have read all of the papers you suggested and have decided to cite ‘four’ articles out of them to support our statements in the introduction. 

Articles by Lin, C.-Y, et.al, Chang, K.-C. et. al, are cited along with Karow A. et.al in line 106, page 5 and article by Cheng, C.-M., et. Chang, K.-C. et.al are cited along with Couto E Cruz C et. al, and Singh S. et al in line 108, page 5.

As reviewer 2, advised to reduce the size of introduction, we decided not to add additional text but only contextualize references which supports the statements made in the introduction through line 103-108, page 5.

Comment 3: When introduction the WHOQOL-BREF (i.e., line 174), please also mention that the WHOQOL-BREF has been validated among heroin users.

Chang, K.-C., Wang, J.-D., Tang, H.-P., Cheng, C.-M., & Lin, C.-Y. (2014). Psychometric evaluation using Rasch analysis of the WHOQOL-BREF in heroin-dependent people undergoing methadone maintenance treatment: further item validation. Health and Quality of Life Outcomes, 12, 148.

Our response: Thank you for this very valuable input. We accept your suggestion and have added this reference in line 189-190, page 8. 

The WHOQOL-BREF has been validated among OAT service users and translated into Nepali language and used in research in Nepal.

Comment 4: 4. Please do not use p=<0.001. This presentation is confusing as no one know if it means p=0.001 or p<0.001.

Our response: Thank you for pointing out this issue. It was a typing error and it actually is p<0.001, which can now be seen corrected in Table 2. 

Comment 5: The Results section is lengthy and has redundancy between text and tables. The authors are suggested to report concise results in text and refer the readers to read the details in tables.

Our response: Thank you for your comment. We have formatted the result section to reduce redundancy and highlighted the major findings from each table. We have removed repetition of numbers from tables and referred to respective tables where ever appropriate. You can follow the changes in result section from line 241-255, 260-266, 276-287, 296-304 and 311-329 of through pages 10-20 (Results section) in the “revised manuscript with track changes.”

Comment 6: The authors have defined QoL as quality of life. Then, they should adhere to using this abbreviation. However, the authors sometimes still use quality of life (e.g., line 316; quality of life among OAT service users).

Our response:

Thank you for your important comment. We have made changes in line 100 (page 5), line167 (page 7), line 188 (page 8), line 337 (page23), line 397 (page 25), line 431 and 438 in Page 27

Comment 7: The Discussion may discuss the QoL and psychological distress issues among people with OAT are related to stigma among this population. The authors may consult the references I mentioned earlier. The authors may also encourage future studies to know the stigma levels (including perceived stigma and self-stigma) among this population given that valid instruments have been developed (e.g., Chang et al., 2020; Fan et al., 2022).

Chang, C.-C., Chang, K.-C., Hou, W.-L., Yen, C.-F., Lin, C.-Y., & Potenza, M. N. (2020). Measurement invariance and psychometric properties of Perceived Stigma toward People who use Substances (PSPS) among three types of substance use disorders: heroin, amphetamine, and alcohol. Drug and Alcohol Dependence, 216, 108319.

Fan, C.-W., Chang, K.-C., Lee, K.-Y., Yang, W.-C., Pakpour, A. H., Potenza, M. N., & Lin, C.-Y. (2022). Rasch Modeling and Differential Item Functioning of the Self-Stigma Scale-Short Version Among People with three different Psychiatric Disorders. International Journal of Environmental Research and Public Health, 19, 8843.

Our response:

Thank you for your inputs in regards to stigma. We did not discuss in detail about stigma in this paper because our focus of study was psychological distress and QoL. Furthermore, we are working in a follow-up manuscript evaluating stigma among OAT service users; hence it was not much addressed in this paper. However, we have taken your recommendation and made statement for need of further studies to know about perceived and self-stigma among OUD and OAT service users, citing relevant references suggested by you. 

Line 401-405, Page 25-26 reads as:

Further research might explore potentially direct and indirect impacts of a broader range of social and structural variables that appear to be associated with QoL and SPD, such as the role of perceived stigma and self-stigma. This research would benefit from quantitative, qualitative, longitudinal, and intervention studies.

Reviewer #2:

Comment 1

- In the title, emphasize that it is about drug use when it comes to the division of participants into injecting and non-injecting “Psychological distress and quality of life among Opioid Agonist Treatment service users with a history of injecting and non-injecting drug use: A cross-sectional study in Kathmandu, Nepal”

Our response: Thank you for your comment. We have addressed the issue as suggested by you. The title (line 1-4, Page 1) now reads as:

“Psychological distress and quality of life among Opioid Agonist Treatment service users with a history of injecting and non-injecting drug use: A cross-sectional study in Kathmandu, Nepal”

Comment 2

In the introduction, the problem is nicely and chronologically explained, but the introduction should be a little more concise or shorter.

Our response:

Thank you for emphasizing the need for making a more concise and shorter introduction. We have a deleted sentences from line 109-113 (page 5) and line 117-119 (page 5). 

The new condensed paragraph now reads as:

In addition to physical and mental health concerns, IDUs often have instability in many aspects of life in addition to high prevalence of infectious disease and mental disorders, such as crime, violence, and lack of stable housing. Resource poor settings often face greater difficulties in addressing the diverse but entangled needs of persons with OUD, due to competing health needs and other priorities.

Comment 3

- In the methodology, it is necessary to briefly explain why you choose 5 out of the 12 sites where the OAT programs were implemented, and whether the programs in those cities differ from the others in terms of content. This is important because of the external validity of the study and the generalizability of the results.

Our response:

Thank you for your comments. Kathmandu valley has three districts – Kathmandu, Lalitpur and Bhaktapur. Kathmandu valley has a total of 5 OAT centers and each of these sites were included in the study. Kathmandu valley was chosen on convenience, as it has 5 out of 12 cites, and caters more than half of all OAT service users from all over the country. The other OAT centers are also situated across major cities (urban area) and the service users are managed by the same guiding protocol and organogram. Hence, we believe that the five centers represent OAT service users from the over country.

We have added the following sentences to address your valid concerns for generalizability and external validity 

Line: 136-137 (page 6)- All OAT sites providing service in Kathmandu were selected for the study.

Line 139-143 (page 6)- Kathmandu valley was chosen on convenience as it has the highest number of OAT sites and highest cumulative OAT service users. The center in Kathmandu represents remaining OAT sites in other cities, as all other centers are also urban based and operating as per the same treatment protocol. 

Comment 4

- It is not recommended to start sentences with abbreviations

Our response:

Thank you for your comments. We have scanned the document in detail and found some places where sentences started with abbreviations. We have addressed the issue as per your suggestion. 

Line 87-88 (page 4) … “OAT programs are”……, has been paraphrased to “The National Center for AIDS and STD Control (NCASC) implements OAT programs at 12 sites”

Line 409 (page 26) …”OAT programs must be”….. has been paraphrased to “Sites of OAT programs…”

Line 431 (page 27) “OAT service users in this…” has been paraphrased to “ Service users in OAT programs in this….”

Comment 5

Have you used the Kolmogorov - Smirnov test to test the normality of the data distribution, in case the data distribution is not normal it is necessary to use non-parametric tests.

Our response:

Thank you for your comment and important observation. When we worked on the manuscript, test for normality was based on histogram observation and examination using skewness and kurtosis. 

In a manuscript by Kim HY, 2013, the author mentions for medium sized samples (50<n<300), the null hypothesis is rejected at an absolute z-value +/- 3.29. This was the basis of our testing for Normal distribution. From this test, all the variables were deemed to be normally distributed hence we progressed with the parametric test. 

• Kim HY. Statistical notes for clinical researchers: assessing normal distribution (2) using skewness and kurtosis. Restor Dent Endod. 2013 Feb;38(1):52 54. 

https://doi.org/10.5395/rde.2013.38.1.52)

From the manuscript

“For medium-sized samples (50 < n < 300), reject the null hypothesis at absolute z-value over 3.29, which corresponds with a alpha level 0.05, and conclude the distribution of the sample is non-normal.”

In recent times, other interesting publications have also highlighted using parametric test for non- normal distribution in medical research. We want to highlight two publications which has taken this discussion forward:

• Cessie S le, Goeman JJ, Dekkers OM. Who is afraid of non-normal data? Choosing between parametric and non-parametric tests. Eur J Endocrinol. 2020 Feb 1;182(2):E1–3. 

In this manuscript the authors mention,

“In many papers the Methods’ section reads like: ‘for non-normally distributed data, non-parametric tests were used’. And indeed, many papers apply non-parametric tests, such as Mann–Whitney test or Wilcoxon test, to compare groups, when the data do not seem completely normally distributed. However, the use of parametric methods, like the t-test, has a clear advantage compared to non-parametric tests: where a non-parametric test will only produce a P value, a t-test will also produce the observed mean difference between the groups, with a 95% confidence interval (CI)…….”

• Wadgave U, Ravindra Khairnar M. Parametric test for non-normally distributed continuous data: For and against. Electron Physician. 2019 Feb 25;11(2):7468–70.

In this manuscript authors mention,

“The existing evidence from simulation studies suggests that parametric methods are preferred over non-parametric in most situations while analysing non-normally distributed continuous data. Even though non-parametric tests are independent of normality assumption, they depend on equal shape and variance of the two distributions [homoscedasticity] (12). So, non-parametric tests should only be considered for the continuous data when the distribution is highly skewed and log transformation cannot change it to normal distribution and when normality of these data cannot be assumed from reports of these data elsewhere (16). Considering the limitation of normality tests’ application to both large and small sample sizes, it is advised to assess the magnitude of skewness of data distribution with graphical methods…”

However, despite considering the above inferences using references, we welcome and support the comment of the Reviewer # 2 and Associate Editor and reapproached using Kolomogorov- Smirnov test, and decided to use non-parametric test- “The Mann-Whitney U-test” for variable “Duration since OAT enrolment (months)” in Table 2 (page 14-15) and “K-6 score” in Table 3 (page 16) since these two variables were heavily skewed.

Also, the following addition has been made in the Statistical analysis section:

Lines 215-217 Page 9: The normal distribution of the continuous variables was checked by using visual inspection, assessment of skewness and kurtosis and Kolmogorov-Smirnov test. 

Lines 219-220 Page 10: The Mann-Whitney U-test was used for not normally distributed continuous data.

Comment 6

- Start the results with a description of the sample, not with the sentence "Table 1 summarizes the socio-demographic characteristics of the OAT service user……". This sentence should be placed after the textual description of the contents of Table 1.

Our response:

Thank you for this important observation. We have amended the result section as per your suggestion. Now all the result section starts with a description of the sample. 

Comment 7

In the tables, it is necessary to express one p-value for comparing the value of one independent categorical variable between injecting and non-injecting drug users, for example, for the variable education we have three p-values instead of one. It is necessary to check that the same error is not repeated in the tables below. P value is expressed to three decimal places, and values less than 0.001, e.g. 0.000 is displayed as 0.001

Our response:

Thank you, for your suggestion. We have changed to one p-value to for one independent categorical variable between injecting and non-injecting drug users. In table 1 (Page 11-13), the p-values for the following variables have been re-analysed and changed: Education (p=0.021), Ethnicity (p=0.752), employment status (p=0.216), marital status (p= 0.004). 

Like you suggested, p value <0.001 has been kept as 0.001, in Table 2 (Page 14-15) (Duration since OAT enrolment), Table 3 (Page 16) (Physical QoL, Overall QoL). 

Comment 8

- In line 231, explain the abbreviation SD.

Our response: 

Thank you, SD stands for standard deviation which is labelled in Table 1. So, its deleted from the text portion in the paragraph. 

Comment 9

- In line 240 the sentence "Regarding SES 70 (61.9%) belonged to upper SES as 241 compared to 54(45.8%) non IDU (p=0.014)" is not clear.

Our response: 

We have clarified the text with simple sentence. 

About one- third of non-IDU had co-morbid medical conditions which was significantly less for IDUs, and had higher SES. (line 253-255, page 11)

Comment 10

A cross-sectional study design does not allow the use of words such as predictor or risk factor because we cannot prove causality, but instead the construct "factor associated with" can be used.

Our response: 

Thank you for your comment. We have changed predictor variables to independent variables (Line 230, page 10, Statistical analysis)

Comment 11

- In line 325 “The possible reasons for the gender differences may be due to higher level of perceived stigma, and barriers in access to health care facilities and treatment including OAT for females”…. Are there other gender differences that could be the cause of the obtained results, maybe differences in employment, income...what do other studies say.

Our response: 

Thank you for this important observation. In our study gender difference was not obvious in socio-demographic variables like employment, income etc. The statement made in line 346-348 Page 23, “The possible reason…..including OAT for females” was mostly to highlight the fact that females drug users don’t access treatment as common as men in settings like Nepal. Even in this study the female population (IDU and non-IDU combined) was just 18 out of 231. 

As suggested by the reviewer, we examined if there were differences in regards various socio-demographic variables in regards to gender, but did not find any. Additionally, we have added a reference from a recent systematic review and meta-analysis (Santo Jr, T, et al, 2022) to support our statement in the discussion

Comment 12

In the discussion, the data from the results are often repeated, the discussion should be based more on the comparison with the results from other studies for a potential explanation of them.

Our response:

Thank you, we have taken that into consideration and taken out results and redundancy from our discussions. 

Academic Editor (Eleni Petkari)

Comment : Both Reviewers find that the work is original and well written in general terms. However, a more concise results section is needed, and the execution of the Kolmogorov Smirnoff test is necessary to decide whether to proceed with parametric or not parametric analyses.

Based on the Reviewers' suggestions, the discussion should also be amended, to provided thorough evidence-based explanations of the results.

Our response: Thank you for your comments. We have made the result more concise with deletion of repetition of findings and statistical inferences already evident on the tables. The discussion has also been amended with removal of redundant repetition from results and addition of five new references to support of discussion and observations. 

About the use of Kolmogorov Smirnoff test and use of parametric or non-parametric test, we have responded to that in detail comment 5, Pages 10-12 in ‘Response to reviewer #2’. 

We have reapproached using Kolmogorov- Smirnov test, and used non-parametric test- The Mann-Whitney U-test was used for variable “Duration since OAT enrollment” in Table 2 and “Kessler-6 score” in Table 3. (See above comment 5 Pages 10-12)

Thank you once again for the opportunity to revise and resubmit our manuscript. I would also want to emphasize that the final ‘manuscript’ meets PLOS ONE’s style requirements, including file naming. We look forward to hearing from you.

---

## [Decision Letter · Decision Letter 1]

24 Jan 2023

Psychological distress and quality of life among Opioid Agonist Treatment service users with a history of injecting and non-injecting drug use: A cross-sectional study in Kathmandu, Nepal

PONE-D-22-23448R1

Dear Dr. Pant,

We’re pleased to inform you that your manuscript has been judged scientifically suitable for publication and will be formally accepted for publication once it meets all outstanding technical requirements.

Kind regards,

Eleni Petkari

Academic Editor

PLOS ONE

Additional Editor Comments (optional):

Dear Authors,

the Reviewers are happy with the modifications you provided to the manuscript. I am please to inform you that your article can be accepted for publication.

Please make sure to address a final comment by Reviewer 2, as follows:Table 2 for the variable "Provision of 'take away' OAT (Last month)" to correct the p value, more precisely to be rounded to 3 decimal places during the proofreading process of the publication.

Reviewers' comments:

Reviewer's Responses to Questions

**Comments to the Author**

1. If the authors have adequately addressed your comments raised in a previous round of review and you feel that this manuscript is now acceptable for publication, you may indicate that here to bypass the “Comments to the Author” section, enter your conflict of interest statement in the “Confidential to Editor” section, and submit your "Accept" recommendation.

Reviewer #1: All comments have been addressed

Reviewer #2: All comments have been addressed

2. Is the manuscript technically sound, and do the data support the conclusions?

Reviewer #1: Yes

Reviewer #2: Yes

3. Has the statistical analysis been performed appropriately and rigorously? 

Reviewer #1: Yes

Reviewer #2: Yes

4. Have the authors made all data underlying the findings in their manuscript fully available?

Reviewer #1: Yes

Reviewer #2: Yes

5. Is the manuscript presented in an intelligible fashion and written in standard English?

Reviewer #1: Yes

Reviewer #2: Yes

6. Review Comments to the Author

Reviewer #1: The authors have taken all my previous comments seriously to revise their contribution. The revised manuscript is much improved and I am happy with it. I have no more comments on it now.

Reviewer #2: The authors corrected everything requested of them and improved the manuscript. It is additionally necessary in Table 2 for the variable "Provision of 'take away' OAT (Last month)" to correct the p value, more precisely to be rounded to 3 decimal places.

7. PLOS authors have the option to publish the peer review history of their article (what does this mean?). If published, this will include your full peer review and any attached files.

Reviewer #1: No

Reviewer #2: No

---

## [Editor Report · Acceptance letter]

26 Jan 2023

PONE-D-22-23448R1 

Psychological distress and quality of life among Opioid Agonist Treatment service users with a history of injecting and non-injecting drug use: A cross-sectional study in Kathmandu, Nepal 

Dear Dr. Pant:

I'm pleased to inform you that your manuscript has been deemed suitable for publication in PLOS ONE. Congratulations! Your manuscript is now with our production department. 

Kind regards, 

on behalf of

Dr. Eleni Petkari 

Academic Editor

PLOS ONE